# Barcode-free next-generation sequencing error validation for ultra-rare variant detection

Huiran Yeom [1], Yonghee Lee [1], Taehoon Ryu[2], Jinsung Noh [1], Amos Chungwon Lee [3], Han-Byoel Lee [4], Eunji Kang[5], Seo Woo Song[1] & Sunghoon Kwon[1,2,3,6]

The advent of next-generation sequencing (NGS) has accelerated biomedical research by enabling the high-throughput analysis of DNA sequences at a very low cost. However, NGS has limitations in detecting rare-frequency variants (< 1%) because of high sequencing errors (> 0.1~1%). NGS errors could be filtered out using molecular barcodes, by comparing read replicates among those with the same barcodes. Accordingly, these barcoding methods require redundant reads of non-target sequences, resulting in high sequencing cost. Here, we present a cost-effective NGS error validation method in a barcode-free manner. By physically extracting and individually amplifying the DNA clones of erroneous reads, we distinguish true variants of frequency > 0.003% from the systematic NGS error and selectively validate NGS error after NGS. We achieve a PCR-induced error rate of $2.5 \times 10^{-6}$ per base per doubling event, using 10 times less sequencing reads compared to those from previous studies.

[1] Department of Electrical and Computer Engineering, Seoul National University, Seoul 08826, Republic of Korea. [2] Department of Molecular and Genetical Engineering, Celemics Inc., 371-17, Gasan-dong, Geumcheon-gu, 08506 Seoul, Republic of Korea. [3] Interdisciplinary Program for Bioengineering, Seoul National University, 08826 Seoul, Republic of Korea. [4] Department of Surgery, Seoul National University College of Medicine, Seoul National University Hospital Biomedical Research Institute, 03080 Seoul, Republic of Korea. [5] Cancer Research Institute, Seoul National University, 03080 Seoul, Republic of Korea. [6] Bio-MAX institute, Seoul National University, 08826 Seoul, Republic of Korea. These authors contributed equally: Huiran Yeom, Yonghee Lee. Correspondence and requests for materials should be addressed to S.K. (email: skwon@snu.ac.kr)

High-throughput next-generation sequencing (NGS) technologies[1] have revolutionized biological research and clinical fields by enabling detection of important genetic variants[2–5]. Especially, analyzing rare somatic variants provides clues towards the exact biological status. For example, detecting rare variants in cancer biology can be important indicators for effective treatment strategies through better understanding of the tumor heterogeneity[6,7] and clonal evolution[8,9]. Similarly, early diagnosis of diseases by drug-resistance or organ transplant rejection requires sensitive NGS analysis with high accuracy, since the ratio of the variant is as little as below 1%[10–12]. However, detection of the rare variants at a frequency below 1% remains challenging because of the high NGS error rate (0.1–1%) (Fig. 1a)[13]. The source of the NGS errors is mostly not from the library preparation but are systematic errors (i.e. misreads during sequencing process), which include phasing noise, invalid signal intensity threshold, signal decay along the increasing cycle, signal cross-talk among DNA clusters, and overlap of emission frequency spectra[14]. These systematic NGS errors are difficult to distinguish from true somatic variants, especially when the somatic variants are rarer than NGS errors.

In order to distinguish true variants from the misreads of NGS systematic error, several methods have been developed depending on molecular barcoding strategies[15–18] or data quality control by bioinformatics algorithm[19,20]. The barcoding approaches use read replicates to filter out randomly occurred misread bases through tagging individual DNA molecules with molecular barcodes[21] and producing a consensus sequence from the read replicates of the same barcode sequence. In other words, the true variant can be detected because the variant is located in the same position within the aligned read replicates. Previous studies have reported that at least 10× depth of sequencing reads is required to construct read replicates for detecting rare genetic variants[22]. However, all sequencing reads must be replicated, regardless of whether the sequencing reads represent rare variants or not (i.e., reads with normal sequence or other non-targeted variants). This in turn results in increasing the sequencing cost over 10×, which can be of great concern in clinical experiments with the large number of patients[23]. Additionally, the reads including the rare variants can be buried among other unnecessary reads due to non-normalized read replicates generated during sample barcoding process[24].

Moreover, the bioinformatics quality control relies on the quality score (Q-score) generated by the NGS system itself, which represents error probability ($P$) considering phasing noise, signal decay, mixed clusters, and cross-talk of control signal in base calling system. The Q-score is described as integer-rounded score, $Q = -10\log_{10}P$ and is referred to as the Phred Q-score[25]. Thus, high-quality data is enriched to reduce NGS error by removing NGS reads of low Q-scores[26]. However, since the Q-score does not completely reflect NGS errors, the threshold value of Q-score should be determined considering a trade-off between erroneous reads trimming and loss of correct data. Therefore, a few important reads including critical variants can be removed during data filtering[27,28]. In the cases where rare mutations need to be observed and analyzed, data loss by quality control would lead to distorted outcomes.

Here, we introduce a barcode-free NGS error validation method without the need for sample barcoding and the data loss during quality control. Following an NGS run, we physically isolated the corresponding DNA clones of the erroneous reads from NGS substrate, amplified the DNA clones individually and read the sequence of the amplified DNA clones through NGS or Sanger sequencing. This approach enables to distinguish rare true variants from the miscalled bases of NGS error at a rate of below

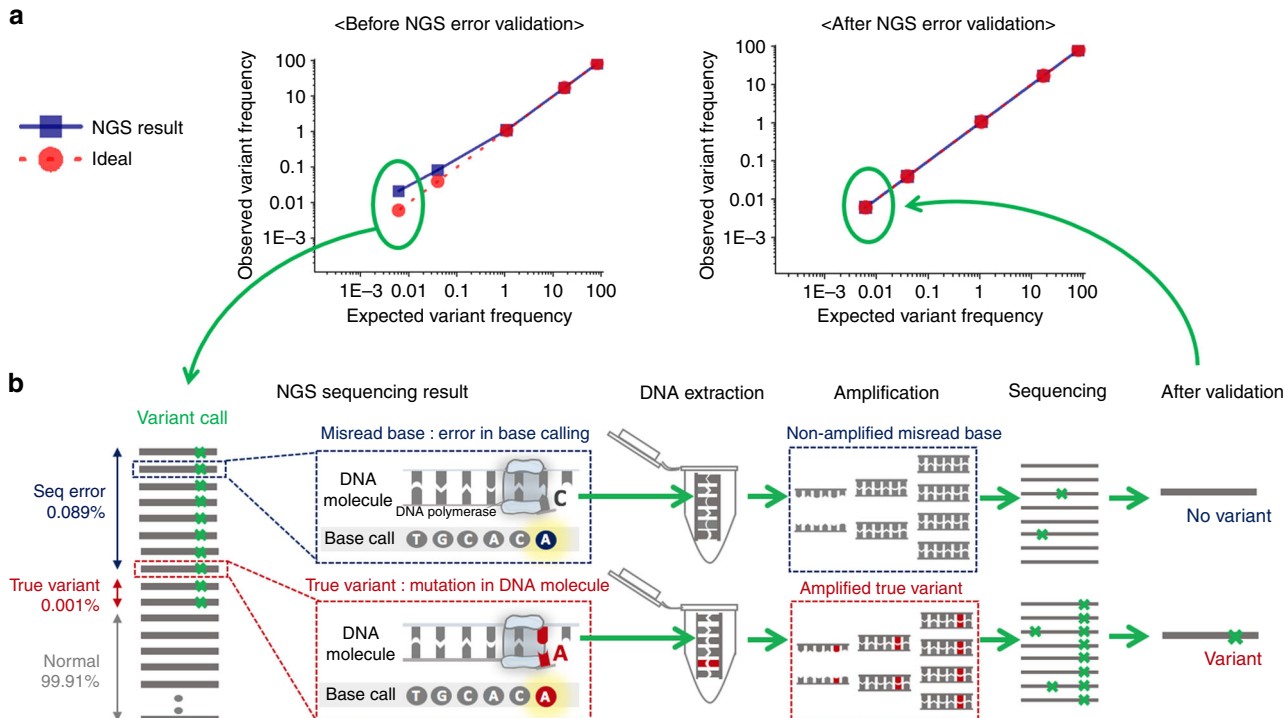

**Fig. 1** The barcode-free NGS error validation method through the DNA clone isolation. **a** The difference of variant frequency (VF) observed before and after NGS error validation. In the NGS base calling system, ~1% of bases are incorrectly identified, which makes it hard to distinguish true variants at a frequency below 0.01%. **b** NGS error validation workflow. The erroneous reads of interest are selected, and their corresponding DNA clones are physically extracted from NGS substrate using LASER isolation. The obtained DNA clones are individually amplified by PCR, whereby only true variants can be duplicated. The amplified DNA clones are sequenced individually. (All data in this figure is based on the real data shown in Supplementary Figure 4.) Source data are provided as a Source Data file

0.1% per base. Using this method, we validate NGS reads of interest selectively in a barcode-free manner, resulting in reduced NGS costs compared to that of molecular barcoding strategies. Additionally, raw NGS data can be used without any filtration by quality control, since any possible erroneous reads can be selected and validated.

## Result

**NGS error validation through selective DNA clone analysis**. For cost-efficient NGS error validation, only erroneous reads of interest should be considered selectively by excluding redundant non-interest NGS reads consumption. The erroneous reads, which are to be determined as variants or NGS errors, can be any reads of interest which need verification, or can be those harboring variations compared to a reference sequence. We approached to analyze specific DNA molecule clones corresponding to the erroneous reads of interest after an NGS run. When the systematic NGS errors are occurred during signal detection, the original molecule remains unchanged. Therefore, we attempted to physically isolate the DNA clones from NGS substrate followed by individual PCR amplification. Since only the true bases can be duplicated, instead of the miscalled bases during PCR, the amplified DNA clones give sequence information that does not contain miscalled bases error in the previous NGS run. We used a laser retrieval system[29] to isolate DNA clones that precisely separates micro-scale objects through radiation pressure of a focused pulse laser at the desired target. For high-throughput isolation, we automated the laser retrieval system which can isolate target DNA clone without human intervention through in-house LabVIEW program (Methods).

The full-process of barcode-free NGS error validation is demonstrated in Fig. 1b. Firstly, erroneous NGS reads of interest were selected as verification targets, which have unintended variations compared to a reference sequence (Methods). Secondly, each DNA clone corresponding to the target reads was extracted from the NGS substrate using the laser retrieval system[29] (Supplementary Figure 1 and Methods) that retrieved over 40 DNA clones per one minute into 96-well plate automatically. Thirdly, the obtained DNA clones were amplified individually by PCR. As the laser retrieval system enables to isolate the DNA clones individually into each well of a 96-well

PCR plate, PCR reaction can be performed right after the retrieval of the DNA clones. Also, we were able to track the corresponding NGS read information through the well location of each selected DNA clone. Finally, the amplified DNA were sequenced individually resulting in the duplicated true bases to be above 95% in the amplified molecules, the removal of NGS error of miscalled bases, and identification of true variants. We sequenced the DNA molecules by Illumina sequencing or Sanger sequencing in those cases where the number of targets was low (<10). This method can also filter out variants, which can be damage, degradation or PCR error of DNA on the NGS substrate, occurred during the validation process (Supplementary Note 1).

**NGS error verification with sequence-known DNA sequencing**. To verify the specificity in distinguishing true variants from the miscalled base errors, we prepared a monoclonal DNA sample of a known sequence (Methods and Supplementary Figure 2). For library construction of the sequence-known DNA samples, we considered minimizing the variants in the DNA molecules by targeting an essential gene of *Escherichia coli* MG1655 (*dapA*) which is known to harbor mutations rarely[30]. We amplified the target gene region (261 bp) through colony PCR and each DNA strand of the PCR product was cloned separately through the Vaccinia DNA *topoisomerase* I cloning method. Additionally, we extracted plasmids from the clones and confirmed their sequences through Sanger sequencing. With this sequence-verified DNA samples, we performed sequencing through 454 junior GS sequencing and selected target reads that have variants to the known sequence (Methods and Supplementary Table 1).

In the NGS result, 15,126 bases (0.147%) and 15,024 bases (0.148%) were indel and substitution bases, respectively, which can be expected as miscalled bases of NGS error. We statistically calculated sample size representing to verify whether the variant calls are true or systematic errors in the NGS result (Methods and Supplementary Note 2). DNA clusters corresponding to 1619 reads (total 160,281 bases) of 817 indels and 1048 substitutions, respectively, were selected (Fig. 2a, Supplementary Data 1). As a result, we confirmed that 99.47% of the variant calls occurred only in the NGS result while there were no variants in the validation sequencing result (Fig. 2b). Notably, all indel variants of 817, except only 1 indel error, were artifact misreads in NGS sequencing. The 1 indel error, which was an insertion of 'C' on

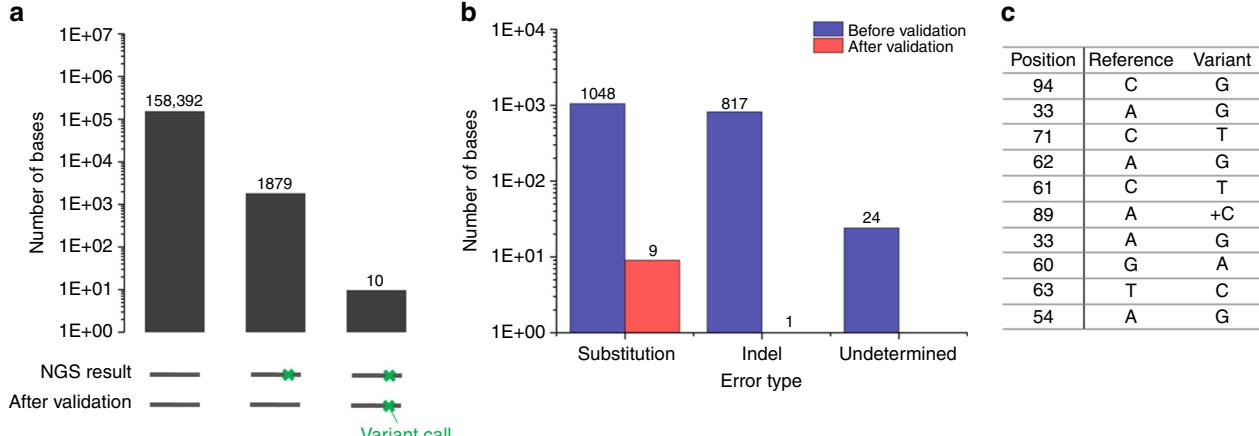

**Fig. 2** NGS error verification of sequence-known DNA sample. **a** Miscalled base errors validation through a known-sequence template sequencing. Out of total variants (1889) called, 99.58% (1879) was miscalled bases, and 0.15% (10) was true variants. **b** Comparison of variants counts before and after the barcode-free NGS error validation according to error type. **c** True variants expected by the barcode-free NGS error validation method; the 10 variants (0.53% of the variant call) were in both of 454 and validation sequencing result that the DNA damage caused. Source data are provided as a Source Data file

the 89th position of the sequence, could have occurred from DNA synthesis error of primer sequence (80–99th position) (Fig. 2c). Additionally, 0.53% of the variant calls were true variants which were true mismatches present in both the 454 and validation sequencing results. We believe that mismatches can be due to DNA damage from sample preparation and storage[31], or contamination caused by mixing DNA molecules of similar sequence.

To establish the sensitivity of barcode-free NGS error validation method, spike-in DNA libraries with different variant fractions of five orders from 0.01% to 90% dilution were used to measure the limit of detection. We assumed that the miscalled bases of NGS errors cause more variants called than the expected variant frequency (VF) in each position. We attempted to verify if the miscalled errors of rare VF (<1%) in DNA samples can be distinguished. In order to distinguish the spike-in DNA samples (0.01–90%) representing each of the VFs in an NGS run, the DNA samples had different variants harboring mutations at different positions. Before NGS run, the DNA samples were quantified by real-time qPCR (Applied Biosystems, 7500 fast) and then diluted from 0.01% to 90% (0.01%, 0.1%, 1%, 10%, and 90%). Additionally, through labeling each of the DNA samples of different variants, we could precisely verify the expected frequency in the mixture after the NGS run as from 0.002% to 95.6% (Supplementary Figure 3, Supplementary Figure 4, and Supplementary Data 2).

In the NGS result, we found the unexpected variants at five positions, from which we obtained a total of 806 reads out of the 164,332 reads in total from four repeats (Supplementary Figure 4 and Supplementary Data 2). Rare variants below 1% of VF were buried by the miscalled bases of NGS error. The sequencing result showed an average of 13.7 times more variants than the expected VF below 1% ($R^2 = 0.77$, <VF 1%). We attempted to verify all the unexpected variants separately for every VF, as shown in Fig. 3a. Through the validation, observed VFs were reduced as the NGS errors were filtered out: 0.053% reduction in VF 90%, 1.2% reduction in VF 10%, 4.5% reduction in VF 1%, 65% reduction in VF 0.1%, and 88% reduction in VF 0.01%. The variant calls in NGS result could be reduced as an average VF to 0.57 times below VF 1% ($R^2 = 0.98$, <VF 1%), resulting in sensitively distinguishing the real variants from NGS error under VF 1% (Fig. 3b and Supplementary Figure 4). Although the detection for sensitivity was limited because of low-throughput reads in 454 sequencing platform (<100,000), we could verify rare variants up to VF 0.003%.

**Distinguishing PCR-induced error from NGS error.** We examined whether this method could distinguish PCR-induced error, which occurs during PCR thermal cycles[32], from NGS error with the lower number of reads (<10 times) than in those from the previous studies[15,22,32]. For constructing the DNA templates, we introduced variations in DNA templates (261 bp) using a prolonged PCR protocol of 60 cycles of PCR resulting in 43 doubling events (Methods and Supplementary Figure 5), resulting in variants with over 0.01% VF accumulated per base. With this DNA sample, an NGS run of 9898 reads including 2,197,356 bases was performed (Methods). Since the PCR-induced error can occur anywhere in a DNA sequence, we extracted all DNA clones with variations in any position compared to the designed sequence (Supplementary Data 3).

Following NGS error validation, we observed the distribution of PCR-induced error along the sequence (Fig. 4a). Additionally, we excluded primer region to avoid counting DNA synthetic error, which can occur during DNA primer synthesis. Our results show that NGS errors occurred more frequently at the end of the sequence and in homopolymer sequences; however, PCR-induced errors occurred randomly (Supplementary Figure 6). In the NGS results, variant calls that most frequently occurred were 'G' insertion errors at the 173rd base position nearby homo-polymer sequence of 'GGG'. However, we confirmed that the 216 insertion errors at this position were artifacts, except for a single variant of substitution, 'G' to 'A'. To analyze the types of PCR-induced error, we selected 1879 substitutions (49.93% of the total substitution error) and 3572 indels (24.97% of the total indels) from the NGS result (Supplementary Note 2). As a result of the verification, there were true variants of 235 substitutions and four indels (Fig. 4b).

Additionally, we wanted to verify if the bases read as error-free in the sequencing results, have a variant. Therefore, we randomly selected 700 DNA clones out of the total 904 error-free reads and extracted them from NGS substrate through laser retrieval (Supplementary Note 2). As a result, we could verify that all DNA clones were error-free with no variants in the DNA molecules. Therefore, with only true variants verified by this method, we calculated the PCR-induced error rate, $2.5 \times 10^{-6}$ per base per doubling event (Methods and Supplementary Figure 5). Comparing with the previous reports[15,22,32], in which the error rates introduced by the same polymerase (Phusion High Fidelity PCR Master Mix, NEB) were measured, the value of the

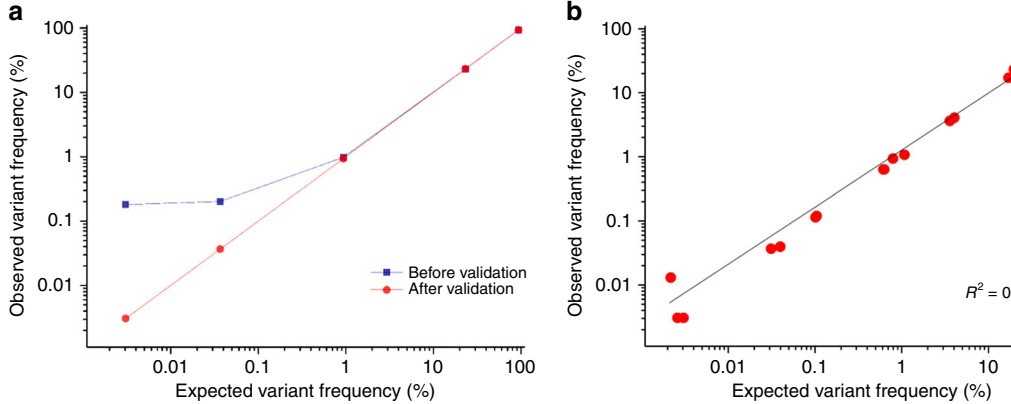

**Fig. 3** The sensitivity of the barcode-free NGS error validation. **a** Barcode-free NGS error validation for detecting spike-in DNA sample varying amounts from 95.6% to 0.002%. Unexpected variants in raw NGS data (Supplementary Figure 3) were verified as NGS error. Accordingly, the $R^2$ values can be calculated using the observed variant frequency (VF) data and expected VF data. With the raw NGS result without NGS error validation, the $R^2$ value was 0.77 at below VF 1%. However, the $R^2$ value was 0.96 after the barcode-free NGS error validation. **b** Unexpected variants were validated as miscalled bases and finally, true variants were distinguished showing varying frequencies, from 95.6% to 0.002%, in four repeats ($R^2 = 0.99$, $n = 19$). Source data are provided as a Source Data file

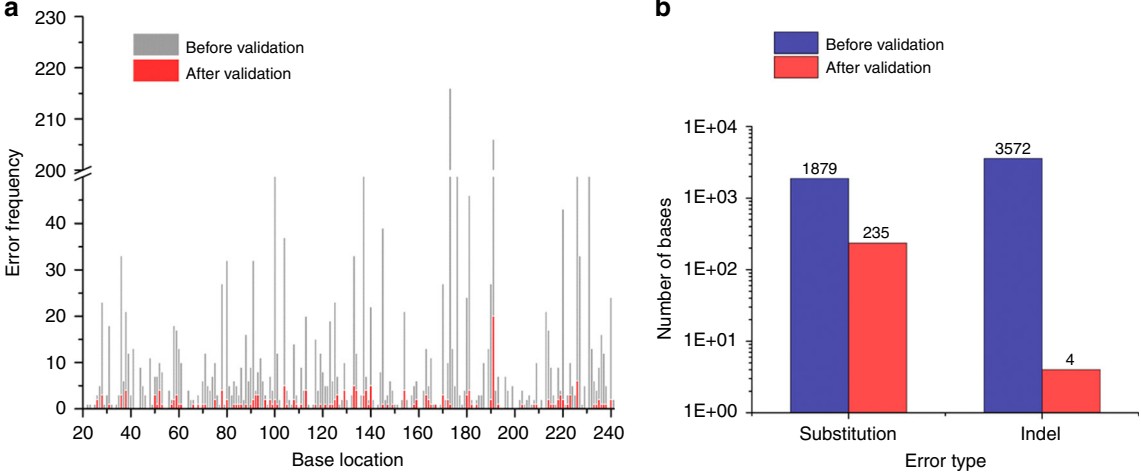

**Fig. 4** Identification of PCR-induced error through the barcode-free NGS error validation. **a** Distribution of NGS and PCR-induced errors in the template sequence. PCR-induced error was randomly distributed while NGS error frequently occurred at homopolymer sequence (e.g. 173rd and 191st base) ($n =$ 2,197,356 bases). **b** Verification of true variants according to error type. 235 of 1879 substitutions and 4 of 3572 indels were true variants (including primer region) after NGS error validation. Source data are provided as a Source Data file

**Table 1 The efficiency of the NGS error validation used to measure PCR-induced error rate**

|  | Conventional[15] | Safe-seq[15] | Potapov, V. et al.[32] | Hestand, M.S. et al.[22] | Barcode-free NGS error validation |
|---|---|---|---|---|---|
| Total sequenced bases | 996,855,791 | 996,855,791 | 118,262,939 | >2,322,766,800 | 2,197,356 |
| Read replicates | – | 1595 | 15 | >10 | – |
| Identified mutations | 198,638 | 197 | 30 | 434 | 202 |
| Error rate (errors/base/doubling) | 9.10E−06 | 4.50E−07 | 3.90E−06 | 1.87E−06 | 2.50E−06 |

The barcode-free NGS error validation can use NGS reads more efficiently at least 10 times compared to the previous reports[15,22,32], since this method does not require read replicates for barcoding. Total sequenced bases represents the bases used to measure variants occurred during PCR with Phusion polymerase. Reads replicates represents the number of read replicates used to filter out miscalled NGS error per a barcode. Identified mutations represents true variants identified by excluding NGS error. Error rate represents the frequency of PCR-induced errors per base per doubling. Source data are provided as a Source Data file

calculated error rate was correlated. In the other methods[15,22,32] to measure PCR-induced error, the read family was required to have more than 10 reads for generating a consensus sequence and filtering out NGS error. However, our method could directly validate NGS error from raw data following an NGS run, making it at least 10 times more efficient in reducing the number of reads required (Table 1).

**Verification of true variants trimmed by quality control**. To check whether raw data quality control can remove not only NGS errors but also true variants of interest, we observed the variants filtered by the barcode-free NGS error validation according to the Q-scores over 10, 20, and 30. We used the NGS result of the PCR-induced error prepared by three kinds of polymerases (Phusion, KAPA, and Q5 DNA polymerase), which have true substitutions over 0.01% of frequency per base (Methods). The NGS result was filtered through a quality filter of FASTX-toolkit, which trimmed each NGS read of average Q-score under 10, 20, and 30. We counted the filtered total reads and variant calls and validated how much true variants can be trimmed through our barcode-free NGS validation method. As a result of Phusion polymerase, ~60.2% of the true variants obtained for >Q10 were excluded when filtered using the highest quality threshold (>Q30); i.e., only 99 variants out of 249 true variants were identified (Fig. 5a). Additionally, in the case of KAPA and Q5 polymerase, the true variants were trimmed as much as 36.2% and 14.2%, respectively (Supplementary Figure 7).

For detailed observation of quality control effects, we examined the number of real variants as the quality threshold increased. The quality control was applied with a 'p 50' option, which means that sequencing reads will be taken if 50% of bases have the quality score over the quality threshold. The examination confirmed that the true variants began to decrease when the filtering Q-score threshold was 18 and decreased the most when the score was 24 (Fig. 5b). These results indicate that quality control by Q-score can result in losing rare variants, especially for >Q20 (Supplementary Figure 7). Furthermore, given that a 'p 50' option is not usually a choice adopted for filtering low-quality reads, there will be more data loss during usual quality control situations where 'p 100' option is applied.

**Discussion**

In summary, we developed a platform to directly examine NGS errors of miscalled bases from NGS raw data, without barcode sequencing and quality control data processing. In this method, we verified that the true variant (>0.003% of VF) can be distinguished from the NGS error. Additionally, we characterized PCR-induced errors, ($2.5 \times 10^{-6}$ per base per doubling), which have been buried by NGS error (~1% per base), with at least 10 times lower than the number of sequenced bases used in the previous studies[15,22,32]. This method avoids extra NGS sample preparation for distinguishing NGS errors from real variants, which could lead to DNA sample loss during the additional steps, such as barcode addition and DNA purification. Additionally, our

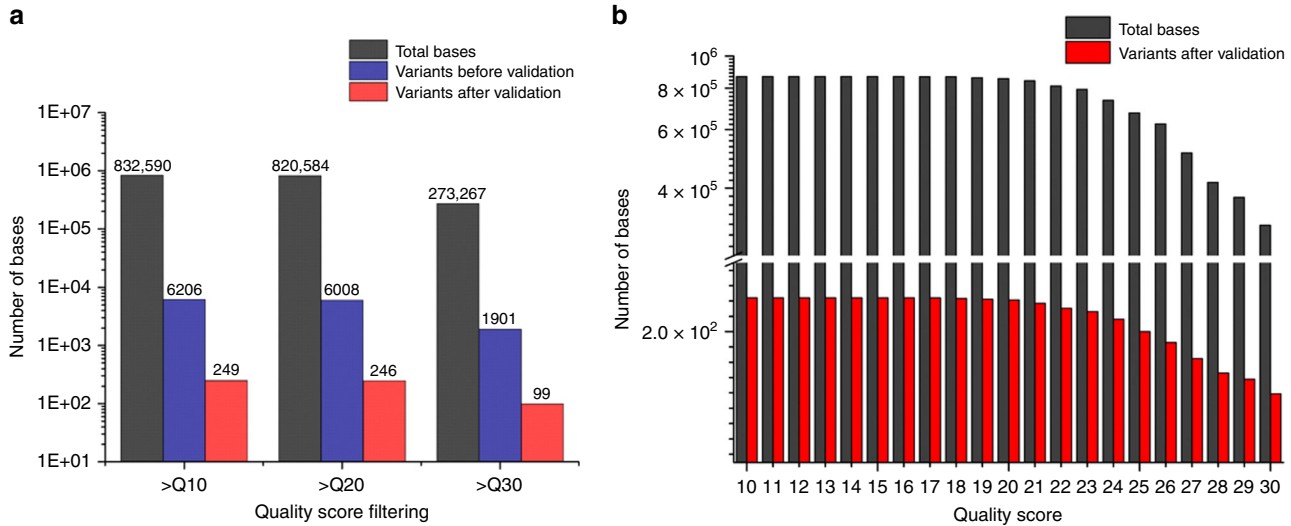

**Fig. 5** Identification of true variants reduction by quality control through the barcode-free NGS error validation. **a** Comparison of variants after filtering raw data with the Q-score (>Q10, >Q20, and >Q30). Variants before the NGS error validation were reduced (3.26 times less) more than variants after validation (2.48 times less), which might include low Q-score of variant calls. **b** Reduction of true variants by quality control from >Q10 to >Q30. From Q-score over than 18, the true variants were confirmed to decrease by the barcode-free NGS error validation. Source data are provided as a Source Data file

method enables to utilize the whole raw NGS data, without quality control filtering, thereby allowing the detection of ultra-rare variants by preserving information of rare variant DNA copies from original sample[27,28]. Since this method can be performed optionally following an NGS run with selective reads validation, this enables selective verification of a few NGS errors, resulting in cost reduction.

However, the number of variant sites to be analyzed and the number of reads containing the target sites are important factors in determining the practicality of this method because the cost of validation sequencing is proportional to the number of target rare variant sites for validation, and inversely proportional to the NGS error rate. In that manner, our method will be more effective in cases where there are few variant sites with rare frequency rather than those with a large number of variant sites. For example, our platform will be effective in applications for quantifying allele fraction in a few variant sites with rare frequency. Specifically, when compared to barcoding methods, our method has cost efficiency when the number of target variant site is lower than ~10,000 sites in single round of analysis, if the NGS error rate is 0.1% in the state-of-art technologies[1] and the depth of the barcoding sequencing is 10 (it is normally done with depth > 10)[15]. Also, if the NGS error rate decreases in the future, our method will be more advantageous for verifying more variants. Therefore, our method could be utilized in studying the low frequency, ultra-rare variants, such as hotspot mutations in circulating tumor DNA or highly diverse sample.

Our method was demonstrated using one specific type of NGS platform, but the fundamental principle of verifying sequencing errors by isolating physical DNA from the NGS-sequencing substrate can be applied to other types of NGS platforms because the fundamental cause of the NGS error in both types of sequencing methods (i.e. sequencing by synthesis and sequencing by ligation) occurs during signal detection itself and is not enzyme-induced (e.g. misincorporation of nucleic acids or damage during signal detection of sequencing process). Proper optimization of isolation technique, such as laser spot size optimization is required for accurate isolation of DNA clusters in the Illumina platform that are more densely packed than those in NGS platform in our demonstration.

We have demonstrated a principle of ultra rare variant detection through analyzing the physical isolated DNA clones from the NGS substrate after the sequencing procedure. Through implementing this idea on more advanced optical or mechanical system, our platform will have impact on wide range of biological and clinical applications in discovering neglected variants that are buried because of the high error rate of NGS.

## Methods

**Library construction.** For preparing monoclonal DNA samples of known sequence, plasmids were extracted from monoclonal *E. coli* clones followed by PCR amplification (95 °C for 2 min followed by six cycles of 98 °C for 30 s, 62 °C for 15 s, 72 °C for 30 s, and final elongation at 72 °C for 2 min) with KAPA HiFi HotStart ReadyMix (KAPA Biosystems). For preparing DNA templates to accumulate PCR-induced error, we extracted *E. coli* genomic DNA by using DNeasy blood & tissue kit (Qiagen), and performed 60 cycles of PCR with the *E. coli* genomic DNA (Supplementary Figure 5). The PCR protocol was according to standard PCR protocol of Phusion® High-Fidelity DNA Polymerase (M0530).

**NGS and quality control.** NGS was conducted by 454 junior GS sequencing (100 cycles) according to the protocols of GS Junior from Roche 454 Life Sciences, 'emPCR Amplification Method Manual—Lib-L'. Also, we used a quality filter of FASTX-toolkit for trimming low-quality reads (Q-score from 10 to 30).

**NGS reads selection for verifying true variants.** Prior to selecting sequencing reads that needed to be validated, we constructed a hash table that mapped *XY* coordinates in 454 junior GS sequencing reads to pixel coordinates in the NGS chip image[16]. The sequencing data was aligned to design sequence using basic local alignment search tool (BLAST) standalone version (BLAST-2.3.0+, NCBI). For verifying true variants of interest, we extracted the information of all sequencing reads that had variant(s) (e.g. substitution, insertion, or deletion) (Supplementary Data 1, 3) or a few sequencing reads that had variant(s) at the desired position (Supplementary Data 2) from BLAST results. These extraction processes were done by the in-house python code. With the hash table, we constructed the list of pixel coordinates of each selected reads. The pixel coordinates were used as positional information for laser retrieval system.

**Laser retrieval system for DNA cluster isolation.** To extract DNA clones physically from NGS substrate, we used laser retrieval system[16], which include Pulse laser (Q-Switched Nd:Yag laser, Minilite, Continuum), true-color charge-coupled device (CCD) camera (Guppy PRO F-146C, ALLIED), two motorized stages, and one inverted microscope (IX71, Olympus) with a ×10 objective lens. Also, we automated to rigorously isolate target DNA cluster without human intervention through in-house LabVIEW program. For automated laser retrieval system, the exact location of the DNA clone on the NGS plate should be calculated to isolate

accurately. Therefore, we approached with two computational methods by considering shorter processing time. First, we developed an image stitching method, which recognized the features on the NGS plate and detected the corresponding center with the decimal value coordinate rather than the integer. Since the offset between different images was not approximated to an integer, the error was not accumulated even if a lot of images (i.e. hundreds) are stitched along one axis. Then, we developed an analytic 'diffusion-like mapping' to calculate the transformation matrix by applying a point pattern matching algorithms, such as invariant to translations, rotations, and scale changes. In order to calculate the location of the desired particles immediately, the matrix is analytically derived from the least-square error estimation of multiple two-dimensional points. Therefore, the exact location of the DNA clones of interest was obtained with high accuracy and in a short time. Over 2500 DNA clusters were retrieved per one hour into 96-well or 384-well plates. And each retrieved beads were amplified separately through PCR conditions of initial denaturation at 95 ℃ for 3 min followed by 26 cycles of 95 ℃ for 30 s, 64 ℃ for 15 s, 72 ℃ for 30 s, and final elongation at 72 ℃ for 5 min with Taq polymerase 2x pre-mix (BioFact).

**Validation sequencing**. Validation sequencing was performed by Illumina Miseq (Celemics, Korea) or Sanger sequencing (Macrogen, Korea). For comparing variants before and after direct NGS error validation, each sequencing reads were aligned to design sequence (*dapA* gene of *E. coli*) using BLAST or Burrows–Wheeler Aligner (BWA) mem aligner (http://sourceforge.net/projects/bio-bwa/files/) followed by processing with SAMtools; view, sort, and mpileup (http://www.htslib.org/doc/samtools.html).

For calling variants, we used VasrScan; pileup2csn (http://varscan.sourceforge.net/using-varscan.html). Finally, each sequencing variants (>80–95% of consensus reads) were compared excluding low reads (>2% of average depth) from Illumina sequencing results.

**PCR-induced error rate calculation**. PCR-induced error (per base per doublings) was calculated as $\frac{\text{True variants}}{\text{Total sequence length}} \div$ doublings. For true variants, we counted the bases according to variants validated through this barcode-free NGS error validation method. For total sequence length, we counted all bases sequenced in 454 sequencing result but the primer region was excluded to avoid DNA synthetic error. For measuring doublings, we quantified gDNA copies before and after PCR amplification through real-time qPCR (Applied Biosystems, 7500 fast) and divided the amplified DNA copies measured after PCR amplification by the initial DNA copies (Supplementary Figure 5). PCR mixture for qPCR was followed as before PCR amplification: gel-purified *E. coli* gDNA (see in Methods—Library construction) 1 μl, 10 μM, forward primer 1 μl, 10 μM, reverse primer 1 μl, KAPA SYBR FAST qPCR Master Mix (2×) 10 μl, nuclease-free water up to 20 μl. After PCR amplification: the amplified DNA sample after three steps of 60 cycles PCR 1 μl, 10 μM, forward primer 1 μl, 10 μM, reverse primer 1 μl, KAPA SYBR FAST qPCR Master Mix (2×) 10 μl, nuclease-free water up to 20 μl. Primer sequences can be found in Supplementary Table 1.

**Reporting summary**. Further information on experimental design is available in the Nature Research Reporting Summary linked to this article.

**Code availability**. The codes that were used for the research are available using a GitHub repository link provided below. (https://github.com/yonghee91/NGS_error_validation.git)

## Data availability

All sequencing data are available in Sequence Read Archive (SRA) under accession numbers SRR8371843 and SRR8371842. The source data underlying Figs. 1a, 2a–c, 3a and b, and 4a and b and Table 1 and Supplementary Figures 2a, 3b and c, 4, 5a–d, 6 and 7 are provided as a Source Data file. All other data are available from the authors upon reasonable request.

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

## Acknowledgements

This research was supported by Global Research Development Center Program through the National Research Foundation of Korea (NRF) funded by the Ministry of Science and ICT (MSIT) (2015K1A4A3047345). This research was supported by a grant of the Korea Health Technology R&D Project of the Korea Health Industry Development Institute (KHIDI) funded by the Ministry of Health & Welfare, Republic of Korea (grant number: HI18C2282) and the Bio & Medical Technology Development Program of the National Research Foundation (NRF) funded by the Korean government (MSIT) (2018M3A9D7079488). This work was supported by the Brain Korea 21 Plus Project in 2018. We acknowledge H. Lee, S. Kim, and H. Kim for the experimental advice.

## Author contributions

H.Y. and Y.L. designed the study with input from T.R. and performed all experiments. T.R. and J.N. established automation of the DNA extraction system through LabVIEW programming. A.C.L., H.-B.L., E.K. and S.W.S. provided conceptual idea which is possible to apply to clinical field, such as circulating tumor DNA. H.Y. and Y.L. interpreted all experimental data and sequencing result, and wrote the manuscript with input from all authors.

## Additional information

**Competing interests:** S.K., H.Y., Y.L., T.R. and J.N. are authors of a patent application for the method described in this paper (Method for identifying errors occurred by massively parallel sequencing and an apparatus for the same Method for identifying errors occurred by massively parallel sequencing and an apparatus for the same, KR2017011929SA, 2016.04.15). The remaining authors declare no competing interests.

