## [Peer Review File · Nature Communications]

Reviewers' Comments:

Reviewer #1:

Remarks to the Author:

In the manuscript, entitled "Barcode-free next-generation sequencing (NGS) error validation for ultra-rare variant detection", Yeom et al developed an approach to validate candidate variants with low allelic fraction. This is an interesting area of NGS-related technology development, especially useful for discovering somatic mosaicism.

Yeom's approach includes the following steps. First, perform standard sequencing; second, computationally identify reads carrying possible variants of low allelic fraction; third, using a laser retrieval system to physically isolate NGS DNA clones that carry the possible variant; finally, PCR and re-sequence the DNA clones.

The manuscript is well-written and overall clear with a good illustration of the principles of the method in Fig. 1. I have only one general issues. Although the concept of this approach can be applied to other studies, the exact procedure described in the manuscript may not be widely applied for two reasons described below. This is the major constraint of the study.

The sequencing and capture of DNA clones were tested on a 454 sequencing platform. However, most of NGS nowadays is done on the Illumina HiSeq or NovaSeq platform. Can this procedure be applied to the Illumina platforms and/or other platforms (e.g., Ion Torrent), and will the accuracy be the same?

Given that current NGS data sets are often huge, e.g., 100GB to TB level for a study, step 2 can take weeks. Can the arrays used in NGS, e.g., flow cells in case of Illumina, be preserved for such a long time for the isolation of DNA clones? Will the accuracy be affected?

Reviewer #2:

Remarks to the Author:

Summary:

In this manuscript by Yeom and colleagues, titled "Barcode-free next generation sequencing error validation for ultra-rare variant detection", the authors describe an approach to distinguish true variants with low variant fractions from errors that result during sequencing or amplification of DNA using PCR. This approach involves the physical extraction of DNA clusters from the NGS substrate using a laser retrieval system, followed by PCR amplification of this DNA and sequencing. The sequenced DNA clusters is used to validate a true event if the same variant is present or an NGS error if variant is absent. In performance experiments, the authors showed that the approach can distinguish a true variant with a lower limit of detection sensitivity down to 0.003%. The authors then demonstrate the barcode-free approach can help to determine the PCR-induced error rate; however, this approach does not help to correct this. This approach has the potential to address the challenge of detecting low allelic fraction variants in various important applications, including the study of clonal hematopoiesis, tumor heterogeneity, and circulating tumor DNA (highlighted in discussion).

Overall, the premise of this work is intriguing and potentially useful. However, I have a few comments related to the claims, scalability, and lack of discussion on the limitations.

Major comments:

1) The authors claim that this approach can distinguish NGS errors by extracting DNA from NGS substrate. However, the NGS that is described in this manuscript is performed using 454 GS Junior sequencer which is obsolete and discontinued. How will this approach work with other sequencing platforms that is not 454 pyrosequencing? For optimal utility and generalizability to NGS, it would be useful to demonstrate this approach on a more relevant sequencing platform.

2) Please discuss how the size of the DNA molecule will affect the performance of the physical extraction. This can be important because there are different library preparations for genomic DNA and circulating cell-free DNA, which will lead to different molecule sizes.

3) Please discuss and compare if there is a trade-off between validation scenarios:

a) Many variant sites but fewer DNA molecules per site. This may be great for overall, global NGS error rate but potentially limited for users who want a corrected variant allele fraction at specific sites for their applications.

b) Few variant sites but most (or all) DNA molecules at each site. This is ideal for quantifying allele fraction but is limited to few sites.

4) Many applications require accurate, error-corrected quantification of allelic fractions. Please discuss or show the scalability of this approach (i.e. how many variant sites) for determining an accurate allelic fraction?

5) How are early PCR errors accounted for during the amplification of retrieved DNA clusters? Will this lead to false variant calls during validation sequencing?

Minor Comments:

1) Please explain why the "distance from the true value" of 20% NGS error is used in the binomial model?

2) The text is sometimes confusing because of the terminology that is used. For example, on pg4, "erroneous reads" are reads of interest but these should also include reads with true variants and not necessarily errors?

3) Pg 7, "different variant fractions of 5 orders from 0.001% to 90%" and "diluted to make VF from 0.01% to 90%" – these numbers do not match up. Then, in Figure 3, it's 0.001% to 99%.

4) In the DNA spike-in experiment, how many of the 806 reads are in each replicate and each dilution? How many DNA clones were retrieved for each replicate and each dilution? This speaks to the sample size for each replicate and dilution when reporting the error filtering.

5) Figure 1a, are these plots based on real data or is this a hypothetical example? Please clarify in the figure caption.

6) References 15, 17, 18, and 27 do not contain journal names.

Reviewer #1

- ✓ ***Comment (1)*** Can this procedure be applied to the Illumina platforms and/or other platforms (e.g., Ion Torrent), and will the accuracy be the same?

Our response:

This is the most important and frequently asked question. Our answer is YES. Our method is applicable to other NGS platforms with similar accuracy shown in this study.

Our method to verify the systematic NGS errors is, intrinsically speaking, analyzing the physically isolated DNA clones from the NGS substrate after the sequencing procedure. The key strategy stems from the observation that the systematic NGS error is caused during the signal detection process, and the enzyme-induced error (e.g. misincorporation of nucleic acids or damage during sequencing process) can be filtered out. Therefore, that the systematic NGS error causing mechanisms should be the same for different NGS platforms is an important factor in considering applicability of our method to other NGS platforms. After confirming that what we are analyzing is common for all NGS, it is important to determine technical applicability of the laser-based isolation platform to be used in other NGS platforms. For technical implementation, it is necessary to physically separate DNA molecules from the NGS substrates that are different for each and every NGS platform. Accordingly, to show that this method is applicable to other NGS platforms, we assessed the systematic NGS error causing mechanisms of other NGS platforms and the technical feasibility of the laser-based DNA molecule isolation system. To improve on these two points in our manuscript, we added the assessments in the discussion section in the revised version of the manuscript.

We first assessed that our platform is applicable to other NGS platforms since the systematic NGS error causing mechanisms are the same as used in this study. When the systematic errors in other NGS platforms are occurred, the major molecules remain unchanged. This is because the errors occur during signal detection, which includes phasing noise, invalid signal intensity threshold, signal decay along the increasing cycle, signal cross-talk among DNA clusters, and overlap of emission frequency spectra (Laehnemann, D., et al., *Brief. Bioinform.* **17**, 154–179 (2016)). Although the enzyme-induced errors during the sequencing methods (i.e. sequencing by synthesis or sequencing by ligation) changes the sequence of the physical DNAs in molecular clones on the NGS substrate, they can be filtered out using simple computational tools. This is because there is an extremely low possibility of enzyme-induced error dominating and altering the signal at the position of the DNA clusters (Supplementary note 2). According to these reasons, the systematic NGS error in other NGS platforms is caused by the same mechanism, which is signal misdetection in sequencing process, as that in the NGS platform we demonstrated in the manuscript. Therefore, the same principle can be applied to other NGS platforms and the errors can be verified through our approach.

Second, in terms of technical feasibility, it is necessary to determine if the DNA clones can be separated using the optical laser system on the different NGS substrates. Since NGS platforms have diverse and different substrates, we need to optimize the laser retrieval system according to each NGS platform. For example, laser ablation is only applicable to transparent NGS substrates because the laser cannot be focused in the inner part of an opaque substrate. The previous study

in our group has demonstrated isolation of DNA clusters from Illumina sequencing plate, which is transparent (Lee, H. *et al.*, *Nat. Commun.* **2**, 1–7 (2015)). In the previous study, although the laser system could isolate two DNA clusters within a single laser spot (10um), we succeeded in verifying their sequences (Figure R1, Lee, H. *et al.*, *Nat. Commun.* **2**, 1–7 (2015)). In this case, the bottleneck was the large spot size (>10um) of the focused nanosecond laser, and if we use picosecond or femtosecond pulse laser we can reduce the laser spot size. When the spot size is reduced, the accuracy will increase because the accuracy of the NGS error verification depends on the ability of the system to isolate exactly one desired DNA clone from the NGS substrate. In other words, if the optical laser system is able to accurately isolate single DNA clone from the NGS substrate, our platform could be applied to other NGS platforms that use transparent substrates with high accuracy. Therefore, our method can be applied to NGS platforms using transparent substrates such as Illumina's.

Figure R1. Application of laser retrieval to Illumina platform (Miseq); (Left) Pixel locations of significant alignment sequences. (Right) BLAST matching results. (Lee, H. *et al.*, *Nat. Commun.* **2**, 1–7 (2015))

If we develop our platform a little further, it can be implemented as a universal platform regardless of the NGS substrate transparency. In case of Ion torrent, the NGS substrate is made of opaque material and therefore the DNA clusters cannot be directly separated using laser. The platform then must be optimized by other physical isolation strategies such as integrated thermal bubble generators.

[Redacted]

Our modification to the manuscript: We added following text to clarify these issues.

(Page 11 line 21 in the main text)

“Our method was demonstrated using one specific type of NGS platform, but the fundamental principle of verifying sequencing errors by isolating physical DNA from the NGS sequencing substrate can be applied to other types of NGS platforms because the fundamental cause of the NGS error in both types of sequencing methods (i.e. sequencing by synthesis and sequencing by ligation) occurs during signal detection itself and is not enzyme-induced (e.g. misincorporation of nucleic acids or damage during signal detection of sequencing process). Proper optimization of isolation technique such as laser spot size optimization is required for accurate isolation of DNA clusters in the Illumina platform that are more densely packed than those in NGS platform in our demonstration.

We have demonstrated a novel principle of ultra rare variant detection through analyzing the physical isolated DNA clones from the NGS substrate after the sequencing procedure. Through implementing this novel idea on more advanced optical or mechanical system, our platform will have impact on wide range of biological and clinical applications in discovering neglected variants that are buried because of the high error rate of NGS.”

- ✓ **Comment (2)** Given that current NGS data sets are often huge, e.g., 100GB to TB level for a study, step 2 can take weeks. Can the arrays used in NGS, e.g., flow cells in case of Illumina, be preserved for such a long time for the isolation of DNA clones? Will the accuracy be affected?

Our response: Since the DNA molecules in the NGS chips are double-stranded after sequencing, the DNA clusters can be preserved for a long time, especially under proper storage conditions (i.e. 4 °C). Longest time we have tried for storing and resequencing was 8 months without noticeable degradation of the physical DNA. Also, even though some portion of DNAs is damaged or degraded during validation process, it could be filtered out without affecting the accuracy of our error verification method. The DNA clusters are very homogenous since they are amplified clones. For example, each DNA cluster in a flow cell has 1,000 copies of DNA. To affect the accuracy of validation, more than half of DNA molecules (> 500 copies of DNA in Illumina sequencing case) should have the same type of damage or degradation at a same position, of which the probability is extremely low. Therefore, the variants occurred during the validation process can be fully filtered out at the validation sequencing (4th step in our method).

Our modification to the manuscript: We replaced the text with added following text to clarify these issues.

(Page 5 line 20 in the main text)

“This method can also filter out variants, which can be damage, degradation or PCR error of DNA on the NGS substrate, occurred during the validation process (Supplementary note 2)”.

Reviewer #2

- ✓ **Major Comment (1)** The authors claim that this approach can distinguish NGS errors by

extracting DNA from NGS substrate. However, the NGS that is described in this manuscript is performed using 454 GS Junior sequencer which is obsolete and discontinued. How will this approach work with other sequencing platforms that is not 454 pyrosequencing? For optimal utility and generalizability to NGS, it would be useful to demonstrate this approach on a more relevant sequencing platform.

This is the most important and frequently asked question. Our answer is YES. Our method is applicable to other NGS platforms with similar accuracy shown in this study.

Our method to verify the systematic NGS errors is, intrinsically speaking, analyzing the physically isolated DNA clones from the NGS substrate after the sequencing procedure. The key strategy stems from the observation that the systematic NGS error is caused during the signal detection process, and the enzyme-induced error (e.g. misincorporation of nucleic acids or damage during sequencing process) can be filtered out. Therefore, that the systematic NGS error causing mechanisms should be the same for different NGS platforms is an important factor in considering applicability of our method to other NGS platforms. After confirming that what we are analyzing is common for all NGS, it is important to determine technical applicability of the laser-based isolation platform to be used in other NGS platforms. For technical implementation, it is necessary to physically separate DNA molecules from the NGS substrates that are different for each and every NGS platform. Accordingly, to show that this method is applicable to other NGS platforms, we assessed the systematic NGS error causing mechanisms of other NGS platforms and the technical feasibility of the laser-based DNA molecule isolation system. To improve on these two points in our manuscript, we added the assessments in the discussion section in the revised version of the manuscript.

We first assessed that our platform is applicable to other NGS platforms since the systematic NGS error causing mechanisms are the same as used in this study. When the systematic errors in other NGS platforms are occurred, the major molecules remain unchanged. This is because the errors occur during signal detection, which includes phasing noise, invalid signal intensity threshold, signal decay along the increasing cycle, signal cross-talk among DNA clusters, and overlap of emission frequency spectra (Laehnemann, D., et al., *Brief. Bioinform.* **17**, 154–179 (2016)). Although the enzyme-induced errors during the sequencing methods (i.e. sequencing by synthesis or sequencing by ligation) changes the sequence of the physical DNAs in molecular clones on the NGS substrate, they can be filtered out using simple computational tools. This is because there is an extremely low possibility of enzyme-induced error dominating and altering the signal at the position of the DNA clusters (Supplementary note 2). According to these reasons, the systematic NGS error in other NGS platforms is caused by the same mechanism, which is signal misdetection in sequencing process, as that in the NGS platform we demonstrated in the manuscript. Therefore, the same principle can be applied to other NGS platforms and the errors can be verified through our approach.

Second, in terms of technical feasibility, it is necessary to determine if the DNA clones can be separated using the optical laser system on the different NGS substrates. Since NGS platforms have diverse and different substrates, we need to optimize the laser retrieval system according to each NGS platform. For example, laser ablation is only applicable to transparent NGS substrates because the laser cannot be focused in the inner part of an opaque substrate. The previous study in our group has demonstrated isolation of DNA clusters from Illumina sequencing plate, which

is transparent (Lee, H. *et al.*, *Nat. Commun.* **2**, 1–7 (2015)). In the previous study, although the laser system could isolate two DNA clusters within a single laser spot (10um), we succeeded in verifying their sequences (Figure R1, Lee, H. *et al.*, *Nat. Commun.* **2**, 1–7 (2015)). In this case, the bottleneck was the large spot size (>10um) of the focused nanosecond laser, and if we use picosecond or femtosecond pulse laser we can reduce the laser spot size. When the spot size is reduced, the accuracy will increase because the accuracy of the NGS error verification depends on the ability of the system to isolate exactly one desired DNA clone from the NGS substrate. In other words, if the optical laser system is able to accurately isolate single DNA clone from the NGS substrate, our platform could be applied to other NGS platforms that use transparent substrates with high accuracy. Therefore, our method can be applied to NGS platforms using transparent substrates such as Illumina's.

Figure R1. Application of laser retrieval to Illumina platform (Miseq); (Left) Pixel locations of significant alignment sequences. (Right) BLAST matching results. (Lee, H. *et al.*, *Nat. Commun.* **2**, 1–7 (2015))

If we develop our platform a little further, it can be implemented as a universal platform regardless of the NGS substrate transparency. In case of Ion torrent, the NGS substrate is made of opaque material and therefore the DNA clusters cannot be directly separated using laser. The platform then must be optimized by other physical isolation strategies such as integrated thermal bubble generators.

[Redacted]

Our modification to the manuscript: We added following text to clarify these issues.

(Page 11 line 21 in the main text)

“Our method was demonstrated using one specific type of NGS platform, but the fundamental principle of verifying sequencing errors by isolating physical DNA from the NGS sequencing substrate can be applied to other types of NGS platforms because the fundamental cause of the NGS error in both types of sequencing methods (i.e. sequencing by synthesis and sequencing by ligation) occurs during signal detection itself and is not enzyme-induced (e.g. misincorporation of nucleic acids or damage during signal detection of sequencing process). Proper optimization of isolation technique such as laser spot size optimization is required for accurate isolation of DNA clusters in the Illumina platform that are more densely packed than those in NGS platform in our demonstration.

We have demonstrated a novel principle of ultra rare variant detection through analyzing the physical isolated DNA clones from the NGS substrate after the sequencing procedure. Through implementing this novel idea on more advanced optical or mechanical system, our platform will have impact on wide range of biological and clinical applications in discovering neglected variants that are buried because of the high error rate of NGS.”

- ✓ **Comment (2)** Please discuss how the size of the DNA molecule will affect the performance of the physical extraction. This can be important because there are different library preparations for genomic DNA and circulating cell-free DNA, which will lead to different molecule sizes.

Our response: The size of the DNA molecule will not affect the performance of the physical extraction as long as it can be sequenced by the specific sequencer. The physical isolation process using laser is confirmed not to damage the physical DNAs. What the laser is focused and ablate is not the DNA on the substrate but the substrate itself. The microsized part of the substrate is separated and DNAs are attached on the microsized part of the substrate.

The part where the laser is focused is the inner part of the NGS substrate. As depicted in the Figure R3, the laser ablation occurs where the laser is focused, which is set to the inner part of the NGS substrate, and the bead with the DNA clusters is located further below the surface of the substrate. The principle of this phenomenon, and the plasma expansion pressure is the driving force of the laser-based DNA cluster retrieval. (Lee, H. *et al.*, *Nat. Commun.* **2**, 1–7 (2015)) Also, because the wavelength of the laser we used is 532 nm, the green laser pulse itself does not damage the DNA molecules. Therefore, regardless of the DNA molecule sizes, the laser separation of the DNA clusters is a robust process, enabling separation of clonal features with size up to the diffraction limit (~1 μm).

Figure R3. The laser focus dependency of ablational retrieval. Plasma expansion pressure only occurs when the green laser focal spot ($\lambda = 532\text{nm}$) stays inside the substrate. (Lee, H. *et al.*, *Nat. Commun.* **2**, 1–7 (2015))

Our modification to the manuscript: We added the following text to clarify the laser separation process in the Supplementary Fig.1.

(Page 6 line 7 in the supplementary information)

“The size of the DNA molecule will not affect the performance of the physical extraction because the part where the laser is focused is the inner part of the NGS substrate. The laser ablation occurs where the laser is focused, which is set to the inner part of the NGS substrate, and the bead with the DNA clusters is located further below the surface of the substrate”

- ✓ **Comment (3)** Please discuss and compare if there is a trade-off between validation scenarios:
 - a) Many variant sites but fewer DNA molecules per site. This may be great for overall, global NGS error rate but potentially limited for users who want a corrected variant allele fraction at specific sites for their applications.
 - b) Few variant sites but most (or all) DNA molecules at each site. This is ideal for quantifying allele fraction but is limited to few sites.

Our response: Thank you for the important comment. Through this comment, we were able to highlight our advantages over limitations clearer and this manuscript was improved much by addressing the points.

We understand that the reviewer's question is to discuss the NGS error for a given variant site (e.g. hot spot mutation) for the rare variants in the two scenarios. It was also assumed that sequencing was performed at a depth sufficient to measure the allele fraction.

Of the two given scenarios, in case a), our method requires a lot of cost for validating NGS errors in many variant sites, while in case b), our method is more useful to identify the NGS errors with lower cost than the barcoding methods. In other words, as more read numbers are verified, the cost and time increase linearly during our validation method. However, in most cases, rare mutations are buried in the more frequent NGS errors. In this case, the minimum number of reads to be verified will be limited according to the NGS error rate. Therefore, the number of variant sites to be analyzed and the number of reads containing the target sites are important factors in determining the practicality of this method. In that manner, our method will be more effective in cases similar to case b).

Figure R4. Sequence alignment on reference sequence.

Case a) is a situation where there are many variant sites, and it is necessary to verify whether all variant calls generated for each variant site are NGS errors. In this case, if there are fewer DNA

molecules per site (Figure. R4 a), the number of reads to be analyzed absolutely is small, but not smaller than the number of reads with NGS errors. Thus, the number of reads to be verified depends on the number of NGS errors at the sites, resulting in linear increase in cost, according to the number of variant sites.

In case b), only a small number of variant sites can be analyzed, but similarly, the number of reads to be verified depends on the number of absolute variant calls. Therefore, as shown in Figure b, all variant calls (NGS error + variant) generated for a single variant site should be verified, and since the absolute number of variants is large, many reads should be verified. However, considering that this platform is used for rare variants, the number of reads to be verified should still be small.

Considering both cases, the number of variant sites is the main factor determining the cost for NGS error validation and measuring allele fraction.

Our modification to the manuscript: We added this content on the part of ‘Discussion’ to improve the reader’s understanding.

(Page 11 line 7 in the main text)

“However, the number of variant sites to be analyzed and the number of reads containing the target sites are important factors in determining the practicality of this method because the cost of validation sequencing is proportional to the number of target rare variant sites for validation, and inversely proportional to the NGS error rate. In that manner, our method will be more effective in cases where there are few variant sites with rare frequency rather than those with a large number of variant sites. For example, our platform will be effective in applications for quantifying allele fraction in a few variant sites with rare frequency.”

- ✓ **Major Comment (4)** Many applications require accurate, error-corrected quantification of allelic fractions. Please discuss or show the scalability of this approach (i.e. how many variant sites) for determining an accurate allelic fraction?

Our response: Technically, our method has no limit on the number of variant sites analyzed. However, as described in Comment 3, it takes time and cost for the validation process, if the number of variant sites is too large. Therefore, compared to barcoding strategies, our method have advantage in terms of cost when the number of variant site is lower than (depth of the barcoding strategy) / (NGS error rate). For example, if the NGS error rate is 0.1% in the state-of-art technologies (Goodwin, S. et al., *Nat Rev Genet* **17**, 333–351 (2016)), and the depth of the barcoding sequencing is conservatively set to be 10 (it is normally done with depth >10, Kinde, I. et al., *Proc Natl Acad Sci USA*.**108**, (2011)), the number of target variant site should be lower than 10,000 sites. If the NGS error rate decreases in the future, our method will be more advantageous for verifying more variants.

Our modification to the manuscript: We added this content on the part of ‘Discussion’ to improve the reader’s understanding.

(Page 11 line 14 in the main text)

“Specifically, when compared to barcoding methods, our method has cost efficiency when the number of target variant site is lower than approximately 10,000 sites in single round of analysis, if the NGS error rate is 0.1% in the state-of-art technologies¹ and the depth of the barcoding sequencing is conservatively 10 (it is normally done with depth >10)¹⁵. Also, if the NGS error rate decreases in the future, our method will be more advantageous for verifying more variants.”

- ✓ **Major Comment (5)** How are early PCR errors accounted for during the amplification of retrieved DNA clusters? Will this lead to false variant calls during validation sequencing?

Our response: We appreciate the considerate comment. As you mentioned, our method could be affected by the PCR errors during amplification of retrieved DNA clusters. However, the probability of the PCR errors leading to false variant calls is extremely low. The reason is that each DNA cluster is composed of many homogenous DNA molecules. Even if the PCR errors occur during early cycles of amplification, the DNA molecules with the PCR errors will be the small part of a DNA cluster. Because we identified the sequences of the extracted DNA clusters as the most frequently occurring sequences in each position, minor population of DNA molecules (that have PCR errors) can be filtered out. For more detailed explanation, a quantitative calculation is followed.

Before calculating a probability of having false variant calls at the validation step, we'll assume the extreme case; more error-prone conditions. First, assume that the length of the DNA is 400 bp, the number of homogenous DNA molecules in the DNA cluster is 100, and the polymerase used for PCR has a substitution error rate of 10^{-4} / base*cycle. The 400 bp length is quite a long one for the DNA molecules in NGS platform. Also, the 100 is quite a small copy number of each clusters in NGS platform and 10^{-4} / base*cycle is the order of error rate of Taq polymerase, which has a high error rate than other polymerases (Phusion, KAPA, Q5, etc). Second, in the validation step, assume that the sequence of each position is determined as a sequence that occurs more than half of the whole sequences in a DNA cluster. Lastly, assume that the PCR error occurs only at the first cycle of amplification in one specific position of DNA.

Probability of occurring PCR errors in the first cycle is known as followed. (Reiss, Jochen, *et al.*, *Nucleic acids research* 18.4, 973-978 (1990)),

$$P(k) = \beta(nl, k)(cx)^k(1 - cx)^{nl-k},$$

where

k = the number of errors in the first cycle

n = the number of single-stranded copies before amplification

l = the length of DNA molecules

c = proportion of mismatches detected by a given method

x = error rate per base per cycle (error rate of polymerase)

β = coefficient of binomial distribution

Then, the probability of occurring one PCR error from one DNA read is

$$P = \beta(l, 1)(x)^1(1 - x)^{l-1},$$

where k = 1, n = 1, and c = 1.

Therefore, according to our assumption, the probability of leading false variant calls due to the PCR errors in the amplification is followed.

$$P(l, x, n) = \beta(l, 1) \left(\frac{1}{3}\right) (x)^1 (1 - x)^{l-1} 0.5^n,$$

where $l = 400$, $x = 10^{-4} / \text{base} \cdot \text{cycle}$, and $n = 100$.

The $\beta(l, 1)$ is multiplied once because all $0.5n$ DNA molecules must have a same PCR error in a same position. A constant $1/3$ is multiplied because three bases, except a normal base are possible candidates as the PCR error. The power of $0.5n$ means that 50 % of DNA molecules in a DNA cluster have a same PCR error. Applying the value of our assumption, the probability is about $2.4 \cdot 10^{-95}$. Thus, the probability of 50 % of DNA molecules in DNA cluster having the same type of PCR error in first cycle of amplification is extremely low.

As our assumption is an extreme case, the probability will be much lower in real conditions. For example, the length of DNA (l) is usually shorter than 400 bp, polymerase error rate (x) is usually lower than 10^{-4} , and the copy number of DNA molecules in a DNA cluster (n) is larger than 100 in various NGS platforms. Also, the sequence of each position is determined as a sequence that occurs more than 70~90 % of the whole sequences in a DNA cluster.

(Page 3 line 10 in the supplementary information)

Supplementary note 2 in Supplementary information was added.

Supplementary Note 2. The possibility that PCR error leads to false variant calls during validation sequencing

The probability of the PCR errors leading to false variant calls is extremely low. The reason is that each DNA clone is composed of many homogenous DNA molecules. Even if the PCR errors occur during early cycles of amplification, the DNA molecules with the PCR errors will be the small part of a DNA clone. Because we identified the sequences of the extracted DNA clones as the most frequently occurring sequences in each position, minor population of DNA molecules (that have PCR errors) can be filtered out. For more detailed explanation, a quantitative calculation is followed.

Before calculating a probability of having false variant calls at the validation step, we'll assume the extreme case; more error-prone conditions. First, assume that the length of the DNA is 400 bp, the number of homogenous DNA molecules in the DNA clone is 100, and the polymerase used for PCR has a substitution error rate of $10^{-4} / \text{base} \cdot \text{cycle}$. The 400 bp length is quite a long one for the DNA molecules in NGS platform. Also, the 100 is quite a small copy number of each clones in NGS platform and $10^{-4} / \text{base} \cdot \text{cycle}$ is the order of error rate of Taq polymerase, which has a high error rate than other polymerases (Phusion, KAPA, Q5, etc). Second, in the validation step, assume that the sequence of each position is determined as a sequence that occurs more than half of the whole sequences in a DNA clone. Lastly, assume that the PCR error occurs only at the first cycle of amplification in one specific position of DNA.

Probability of occurring PCR errors in the first cycle is known as followed. (Reiss, Jochen, *et al.*, *Nucleic acids research* 18.4, 973-978 (1990)),

$$P(k) = \beta(nl, k) (cx)^k (1 - cx)^{nl-k},$$

where

k = the number of errors in the first cycle

n = the number of single-stranded copies before amplification

l = the length of DNA molecules

c = proportion of mismatches detected by a given method

x = error rate per base per cycle (error rate of polymerase)
 β = coefficient of binomial distribution

Then, the probability of occurring one PCR error from one DNA read is

$$P = \beta(l, 1)(x)^1(1 - x)^{l-1},$$

where $k = 1$, $n = 1$, and $c = 1$.

Therefore, according to our assumption, the probability of leading false variant calls due to the PCR errors in the amplification is followed.

$$P(l, x, n) = \beta(l, 1)\left(\frac{1}{3}(x)^1(1 - x)^{l-1}\right)^{0.5n},$$

where $l = 400$, $x = 10^{-4}$ / base*cycle, and $n = 100$.

The $\beta(l, 1)$ is multiplied once because all $0.5n$ DNA molecules must have a same PCR error in a same position. A constant $1/3$ is multiplied because three bases, except a normal base are possible candidates as the PCR error. The power of $0.5n$ means that 50 % of DNA molecules in a DNA cluster have a same PCR error. Applying the value of our assumption, the probability is about $2.4 * 10^{-95}$. Thus, the probability of 50 % of DNA molecules in DNA cluster having the same type of PCR error in first cycle of amplification is extremely low.

As our assumption is an extreme case, the probability will be much lower in real conditions. For example, the length of DNA (l) is usually shorter than 400 bp, polymerase error rate (x) is usually lower than 10^{-4} , and the copy number of DNA molecules in a DNA clone (n) is larger than 100 in various NGS platforms. Also, the sequence of each position is determined as a sequence that occurs more than 70~90 % of the whole sequences in a DNA clone.

- ✓ **Minor Comment (1)** Please explain why the “distance from the true value” of 20% NGS error is used in the binomial model?

Our response: We set the distance value to 20% to verify the NGS error with a minimum number of samples that we can experiment with. We calculated the minimum sample size that can represent the whole sample, and we confirmed that when the distance value is more than 20% of the true value, the sample size in 4 repeat extractions is enough for us to handle. When we set 20% NGS error, we could calculate the minimum number of bases to be analyzed as 145,858 bases (total 73,176 bases for validating substitution error and total 72,682 bases of indel error, respectively).

Our modification to the manuscript: We revised the content in supplementary note1.

(Page 3 line 2 in the supplementary information)

“Also, we set the distance value to 20% to verify the NGS error with a minimum number of samples that we can experiment with. We calculated the minimum sample size that can represent the whole sample, and we confirmed that when the distance value is more than 20% of the true value, the sample size in 4 repeat extractions is enough for us to handle.”

- ✓ **Minor Comment (2)** The text is sometimes confusing because of the terminology that is used. For example, on pg4, “erroneous reads” are reads of interest but these should also include reads with true variants and not necessarily errors?

Our response: As the reviewer's comment, the term "erroneous reads" is meant to describe the reads of interest to be verified, before the NGS validation. Therefore, the erroneous reads can turn out to be variants or NGS errors. The erroneous reads can be verified as variants or errors after the NGS error validation.

Our modification to the manuscript: We added the following text to clarify the word in main text. Also, in other revisions, we tried to use terminologies more clearly to avoid any confusions.

(Page 4 line 18 in the main text)

"The erroneous reads, which are to be determined as variants or NGS errors, can be any reads of interest which need verification, or can be those harboring variations compared to a reference sequence."

- ✓ **Minor Comment (3)** Pg 7, "different variant fractions of 5 orders from 0.001% to 90%" and "diluted to make VF from 0.01% to 90%" – these numbers do not match up. Then, in Figure 3, it's 0.001% to 99%.

Our response: The phrase "diluted to make VF from 0.01% to 90%" in the manuscript was meant to describe the range we diluted the DNA sample with, and the actual final variant frequency we measured was ranged from 0.002% to 95.6% (Supplementary Figure 3). In the original text, the measured VF value was inaccurately expressed as "0.001% to 90%", and we thought this can be confused with the dilution range of "0.01% to 90%". So we tried to differentiate the measured VF from the dilution range using the accurate number: "from 0.002% to 95.6%". Also, in the revised manuscript, we fixed any grammatical errors that have caused confusions.

Our modification to the manuscript: We revised the number of VF in Figure 3 of the supplementary information, which is about the sample preparation, and main text. Also we revised the manuscript and Figure 3 following reviewer's comment. We highlighted the revised part in the manuscript and supplementary information. Thanks for notification.

- in the main text,

(Page 7 line 2)

"from 0.01 % to 90 % dilution"

(Page 7 line 6)

"(0.01 % ~ 90 %)"

(Page 7 line 9)

then diluted from 0.01 % to 90 % (0.01 %, 0.1 %, 1 %, 10 %, and 90 %).

(Page 7 line 12)

"as from 0.002% to 95.6%"

(Page 20 line 5)

"from 95.6% to 0.002%"

(Page 20 line 11)

"from 95.6% to 0.002%"

- in the supplementary information,

(Page 8 line 8)

“and the range of VF was from 0.002% to 95.6%.”

(Page 9 line 4)

“from 95.6 % down to 0.002 %”

- ✓ **Minor Comment (4)** In the DNA spike-in experiment, how many of the 806 reads are in each replicate and each dilution? How many DNA clones were retrieved for each replicate and each dilution? This speaks to the sample size for each replicate and dilution when reporting the error filtering.

Our response: For measuring the sensitivity of our validation method, We diluted DNA spike-in sample with different variant fractions of 5 orders from VF 0.01 % to 90 % resulting in the corresponding VF measured from 0.002 % to 95.6 %. Also, we retrieved 806 suspicious reads from four DNA substrates (four replicate experiments). The number of reads retrieved from each replicate is 254, 240, 188, and 124 reads, respectively (see supplementary table 3, ID column of each sheet). Some reads have more than one variant calls, so the number of suspicious variants is 819 bases. The distribution of retrieved DNA reads is shown in table 1. There is a tendency of having more suspicious reads under VF 1 %, possibly because the effect of misread errors becomes dominant at the low VF (< 1 %).

Table R1. The number of retrieved DNA reads

	1 st replicate	2 nd replicate	3 rd replicate	4 th replicate
0.01 %	66	52	61	99
0.1 %	92	102	58	13
1 %	26	18	14	9
10 %	63	66	0	0
90 %	11	3	53	3

Our modification to the manuscript: We added this content in the supplementary information as supplementary Figure 4 to improve the reader’s understanding. Also, we added the information with the following text in the main text.

(Page 7 line 13 in the main text)

“(Supplementary Fig.3, Supplementary Fig.4, and Supplementary Table 3.)”

- ✓ **Minor Comment (5)** Figure 1a, are these plots based on real data or is this a hypothetical example? Please clarify in the figure caption.

Our response: In Figure 1a, the plots are based on the real data (4th_repeat graph in Supplementary Figure 4). In Figure 1b, the frequency of “Seq Error”, “True variant”, and “Normal” was arbitrary number, which represent a case for problem of the NGS error. To organize the data more clearly, we revised the manuscript and added a part that all data in Figure 1 was based on the real data.

Our modification to the manuscript: We revised the Figure 1 based on real data, and added a sentence in the figure caption.

(Page 18 line 11 in the main text)

“(All data in this figure is based on the real data shown in Supplementary Figure 4.)”

✓ **Minor Comment (6)** References 15, 17, 18, and 27 do not contain journal names.

Our response & modification to the manuscript: We added the journal names in the references. Thank you for the comment.

Reviewers' Comments:

Reviewer #1:

Remarks to the Author:

I think the author responded well to my comments and this manuscript would be a useful contribution to the literature.

Reviewer #2:

Remarks to the Author:

The authors have addressed all my concerns. Thank you for the detailed and organized responses.